# Bufalin Suppresses Head and Neck Cancer Development by Modulating Immune Responses and Targeting the β-Catenin Signaling Pathway

**DOI:** 10.3390/cancers16152739

**Published:** 2024-08-01

**Authors:** Nour Mhaidly, Noura Barake, Anne Trelcat, Fabrice Journe, Sven Saussez, Géraldine Descamps

**Affiliations:** 1Department of Human Anatomy and Experimental Oncology, Faculty of Medicine, Research Institute for Health Sciences and Technology, University of Mons, Avenue du Champ de Mars, 8, 7000 Mons, Belgium; nour.mhaidly@umons.ac.be (N.M.); noura.barake@student.umons.ac.be (N.B.); anne.trelcat@umons.ac.be (A.T.); sven.saussez@umons.ac.be (S.S.); 2Laboratory of Clinical and Experimental Oncology (LOCE), Institute Jules Bordet, Université Libre de Bruxelles (ULB), 1000 Brussels, Belgium; fabrice.journe@hubruxelles.be

**Keywords:** head and neck cancer, bufalin, apoptosis, cell cycle, EMT, macrophage, TME, M2 Reprogrammation, Wnt/β-catenin, EGFR, STAT3

## Abstract

**Simple Summary:**

Head and neck cancers are aggressive and challenging to treat due to the severe side effects and toxicity of current treatments. Bufalin, a natural compound from the Chinese toad, has shown promise in fighting various cancers but has not been thoroughly studied for head and neck cancers. Our research aims to explore how bufalin works against these specific cancer cells. By using different techniques, we discovered that bufalin could reduce cancer cell growth, induce cell death, and enhance the body’s immune response against tumors. These findings suggest that bufalin could become a new, effective treatment option with potentially fewer side effects for patients with head and neck cancers. This research could pave the way for developing better therapies and improving outcomes for patients facing this difficult disease.

**Abstract:**

Bufalin, a cardiotonic steroid derived from the Chinese toad (Bufo gargarizans), has demonstrated potent anticancer properties across various cancer types, positioning it as a promising therapeutic candidate. However, comprehensive mechanistic studies specific to head and neck cancers have been lacking. Our study aimed to bridge this gap by investigating bufalin’s mechanisms of action in head and neck cancer cells. Using several methods, such as Western blotting, immunofluorescence, and flow cytometry, we observed bufalin’s dose-dependent reduction in cell viability, disruption of cell membrane integrity, and inhibition of colony formation in both HPV-positive and HPV-negative cell lines. Bufalin induces apoptosis through the modulation of apoptosis-related proteins, mitochondrial function, and reactive oxygen species production. It also arrests the cell cycle at the G2/M phase and attenuates cell migration while affecting epithelial–mesenchymal transition markers and targeting pivotal signaling pathways, including Wnt/β-catenin, EGFR, and NF-κB. Additionally, bufalin exerted immunomodulatory effects by polarizing macrophages toward the M1 phenotype, bolstering antitumor immune responses. These findings underscore bufalin’s potential as a multifaceted therapeutic agent against head and neck cancers, targeting essential pathways involved in proliferation, apoptosis, cell cycle regulation, metastasis, and immune modulation. Further research is warranted to validate these mechanisms and optimize bufalin’s clinical application.

## 1. Introduction

Head and neck cancer (HNC) is the seventh most common cancer globally, with over 900,000 new cases and 467,000 deaths reported in 2020 [1]. HNC is characterized by its aggressive nature, leading to high rates of morbidity and mortality [2]. The overall 5-year survival rate for head and neck squamous cell carcinoma remains less than 50%, a clinical situation that has remained unchanged over the past several decades [3]. 

Key risk factors such as alcohol, tobacco consumption, and human papillomavirus (HPV) infection significantly influence disease progression and treatment outcomes [4]. Notably, HPV-positive HNC often presents a more favorable prognosis compared to their HPV-negative counterparts [5]. Additionally, the tumor microenvironment (TME) plays a critical role in tumor development, therapeutic response, and overall prognosis [6]. Studies have unveiled the highly immunosuppressive nature of the TME in most HNC tumors, characterized by extensive infiltration of macrophages [7]. Importantly, the presence of these macrophages, particularly the protumoral M2 macrophages, has been associated with shorter recurrence-free survival (RFS) and overall survival (OS), underscoring their pivotal role in disease progression [8,9,10,11]. Patients with HNC are generally provided with standard treatment options like surgery, radiotherapy, chemotherapy, or a combination of these modalities [12]. Despite these treatments, 40–60% of patients experience recurrence and do not respond to subsequent therapies [4]. Although cetuximab, an FDA-approved drug targeting the epidermal growth factor receptor (EGFR) pathway, as well as nivolumab and pembrolizumab, two monoclonal antibodies targeting PD-1, have shown promise in improving outcomes, there remains an urgent need for more effective and tolerable treatment options [13,14]. 

Bufalin, a cardiotonic steroid derived from the venom glands of the Chinese toad (Bufo gargarizans), has been used in Traditional Chinese Medicine (TCM) for centuries to treat infections and inflammation [15]. In TCM, bufalin has also been employed as a treatment for heart failure, particularly in Asian countries [16]. Its wide range of biological activities includes cardiotonic, anesthetic, blood pressure regulation, and antineoplastic effects [17]. The primary target of bufalin is the Na+/K+ ATPase (NKA) pump, an integral membrane protein located in the cell’s plasma membrane, essential for maintaining ionic and osmotic balances [18].

Recent studies have revealed bufalin’s potent anticancer properties across various cancer types, including lung [19], breast [20], liver [21], prostate [21], and melanoma [22,23]. Bufalin exerts its effects through mechanisms such as inducing cell cycle arrest, promoting apoptosis, and inhibiting differentiation, migration, inflammation, angiogenesis, and metastasis [24]. Additionally, bufalin can lead to reactive oxygen species (ROS) production, further promoting cell death and inhibiting cancer cell proliferation [25,26]. Bufalin’s apoptotic effects are known to be mediated by the mitochondria-dependent pathway, as demonstrated in bladder cancer cells [27], oral cancer cells [28], and human osteosarcoma [29]. Additionally, it disrupts the cell cycle in gastric cancer [30], leukemia [31], bladder carcinoma [32], and colorectal cancer cells [33,34]. Notably, bufalin inhibits EMT in various cancers by modulating markers associated with mesenchymal and epithelial phenotypes [15,35,36,37]. Furthermore, bufalin modulates crucial signaling pathways implicated in poor prognosis in HNC, including JAK/STAT, Wnt/β-catenin, and EGFR pathways [38]. It also remodels the TME by targeting immune cells, specifically by inducing the transformation of M2 macrophages into the M1 phenotype, thereby enhancing the antitumoral immune response [39,40]. However, the precise mechanisms underlying bufalin’s actions within the TME and its impact on HNC progression necessitate further investigation.

In the context of HNC, bufalin holds promise as a therapeutic agent due to its ability to reduce cancer cell proliferation via apoptosis induction. However, the exact mechanisms underlying bufalin’s actions in HNC remain to be fully elucidated. Our study aims to explore anticancer properties of bufalin in HNC cell lines by evaluating the cell cycle regulation, the apoptosis, the modulation of EMT, and the molecular pathways involved in HNC progression to better clarify the molecular mechanisms and the efficacy of this new treatment in HNC.

## 2. Materials and Methods

### 2.1. Cell Culture

The HPV-negative FaDu human HNC cell line (ATCC^®^ HTB-43, Manassas, VA, USA) and the HPV-positive 93VU-147T cell line (gifted from Dr. De Winter, University Medical Center of Amsterdam, Amsterdam, The Netherlands) were cultured in DMEM (Gibco Dulbecco’s Modified Eagle Medium Westburg, The Netherlands), while the HPV-negative Detroit-562 cell line (ATCC^®^ CCL-138, Manassas, VA, USA) was cultured in MEM (Gibco Dulbecco’s Modified Eagle Medium). 

Both media were enriched with 10% heat-inactivated fetal bovine serum (FBS Premium South America, PAN BIOTECH, Aidenbach, Germany), 2% L-glutamine (200 mM, Gibco, Thermo Fisher Scientific, Waltham, MA, USA), and 1% penicillin/streptomycin (10 mg/mL, Gibco, Thermo Fisher Scientific). The cultures were maintained at 37 °C in a humidified atmosphere with 5% CO_2_. The medium was refreshed every 48 h, and cells were subcultured using Trypsin-EDTA (PAN-Biotech, Darmstadt, Germany) once they reached approximately 80% confluence. Regular testing confirmed that the cell lines were free from mycoplasma contamination.

### 2.2. Isolation of Monocytes

Monocytes were isolated from peripheral blood mononuclear cells (PBMCs) derived from the blood of healthy donors, as described previously [41]. Briefly, CD14+ monocytes were separated using CD14+ magnetic microbeads and separation columns (Miltenyi Biotec, Leiden, The Netherlands) with the MiniMACS Separator (Miltenyi Biotec, Leiden, The Netherlands). The isolated primary monocytes were cultured in RPMI medium (Lonza, Basel, Switzerland). To differentiate the monocytes into M2 phenotype macrophages, Macrophage Colony-Stimulating Factor (M-CSF) (VWR, Saint Louis, MO, USA) was added at a concentration of 100 ng/mL for 7 days. Then, bufalin (PhytoLab^®^, Vestenbergsgreuth, Bavaria, Germany) was added for 48 h with a concentration of 40 nM.

### 2.3. Viability Assay

This test is based on crystal violet (Sigma-Aldrich, MO, USA) staining of live cells after a 72-hour treatment. Cells were initially seeded in a 96-well plate (8000 cells/well for FaDu and Detroit and 10,000 cells/well for 93 VU) and allowed to adhere for 24 h. Following this period, the cells were treated with varying concentrations of bufalin ranging from 0 to 80 nM and incubated for 72 h. After incubation, cells were washed with DPBS (Life Technologies, Carlsbad, CA, USA) and fixed with 4% Paraformaldehyde (Sigma-Aldrich, MO, USA) for 15 min. Subsequently, cells were stained with 5% crystal violet (Sigma-Aldrich, MO, USA) for 30 min at room temperature, washed with distilled water, and left to dry for 2 h. To permeabilize the cells, a triton 0.2% solution (Rohm and Haas Co., PA, USA) was added for 90 min under constant agitation. Finally, the absorbance was read at 570 nm using a spectrophotometer VERSA max (Molecular Devices, Sunnyvale, CA, USA). The IC_80_ values were calculated for each cell line: 67 nM for Fadu; 100 nM for 93VU; and 75 nM for Detroit. For each condition, we performed three biological replicates, each with six technical replicates.

### 2.4. Clonogenic Assay

To evaluate colony formation, 500 FaDu cells, 800 Detroit cells, and 1000 93VU cells/well were seeded in a 6-well plate and allowed to adhere for 24 h. Cells were then treated with various concentrations of bufalin for 2 weeks, with the medium being refreshed once during this period. After 14 days, the cells were fixed with 4% paraformaldehyde and stained with 4% crystal violet for 30 min. The crystal violet was then removed, cells were washed three times with distilled water, and the plates were left to dry for 24 h. The total number of colonies was counted under a microscope.

### 2.5. Live Cell Imaging Analysis of Cytotoxicity, Apoptosis, and Oxidative Stress

The IncuCyte live-cell analysis system (Sartorius, Göttingen, Germany) was used to assess the cytotoxicity of bufalin on cancer cells, as well as its effects on apoptosis and oxidative stress. This system detects the fluorescence of reagents coupled with fluorochromes. We utilized three specific reagents: Incucyte^®^ Cytotox Green Dye to measure cytotoxicity; Incucyte^®^ Annexin V Green Dye to measure apoptotic effects; and MitoSOX Red (Thermo Fisher, Waltham, MA, USA) to measure mitochondrial ROS.

Cells were seeded in a 96-well plate (Corning, NY, USA) in a volume of 100 μL, ensuring they reached 30% confluence after 24 h. Then, the medium was replaced, and the cells were treated with a medium containing bufalin at IC_80_ concentration along with the specific dye. Buthionine sulfoximine (BSO) (30 µM) served as a positive control for the oxidative stress assay, while cisplatin (30 µM) was the positive control for the apoptosis evaluation. 

The plate was incubated at 37 °C for 30 min before being placed in the IncuCyte^®^ Live-Cell Analysis System for 72 h. Images were captured every 2 h over 72 h to generate cell growth curves, illustrating fluorescence intensity, which was directly proportional to the number of dead cells (The results were analyzed using the IncuCyte software, Version: 2022B Rev2).

### 2.6. TMRE Staining

TMRE staining was used to detect the mitochondrial membrane potential (ΔΨm). Cells were seeded in an 8-well chamber slide (ibidi, Gräfelfing, Germany) for 24 h. The cells were then treated with bufalin at IC_80_ concentration for 24 h. After this treatment period, the medium was removed, and a solution containing 2 μM TMRE (MedChemExpress^®^, NJ 08852, USA) was added for 30 min at 37 °C. The slide was then rinsed with HBSS, and Hoechst dye (BisBenzimide, Sigma Aldrich, St. Louis, MI, USA) was applied for 15 min to stain nuclei. Cells were rinsed twice with PBS for 5 min each time before being examined under a confocal microscope (Nikon Ti2 A1RHD25, Tokyo, Japan) at 37 °C.

### 2.7. Quantitative RT-PCR

RNA was extracted from cell pellets using the InnuPrep RNA mini kit 2.0 (Annalytik Jena, Jena, Germany), following the manufacturer’s instructions. The concentration and purity of the isolated RNA were measured using a nanodrop device (Bio-Drop μlite, Fisher Scientific, Waltham, MA, USA). For a reverse transcription into cDNA, 1 or 2 μg of RNA was used with the Maxima First Strand cDNA Synthesis Kit for RT-qPCR with dsDNase (Thermo Scientific, K1671, Waltham, MA, USA). The cDNA purity was also assessed with a nanodrop and then diluted tenfold before proceeding with quantitative real-time PCR (qPCR) analysis. The qPCR reaction mixture included SYBR Green Mix (Takyon Rox SYBR Core Kit Blue dTTP, Eurogentec, Selland, Belgium), 10 μM of both forward and reverse primers (provided by IDT, Integrated DNA Technologies, Leuven, Belgium) (see Table 1), and RNAse-free water. This mixture was distributed into a 96-well plate (Microplate for PCR with sealing film, 732–1591 VWR), and cDNAs were added to each well in triplicate. The plate was then placed in a thermocycler (LightCycler 96 FW13083, Roche, Bâle, Switzerland). The qPCR program included the following steps: an initial denaturation at 95 °C for 10 min, followed by 40 cycles of 15 s at 95 °C and 1 min at 60 °C for amplification. The melting curve analysis involved 15 s at 95 °C, 1 min at 62 °C, and continuous acquisition at 95 °C. Data were analyzed using LightCycler^®^ 96 SW 1.1 software, with gene expression levels normalized to the housekeeping gene 18S using the 2^−ΔCt^ method.

### 2.8. Western Blot

Proteins were extracted using Mammalian Protein Extraction Reagent and Halt Protease Inhibitor Cocktail, EDTA-Free (ThermoFisher Scientific, Rockford, IL, USA). The cells were lysed by sonication, then centrifuged, and the supernatant was collected. Protein concentrations were determined using the Pierce BCA Protein Assay Kit (ThermoFisher Scientific, Rockford, IL, USA). Subsequently, 20–40 µg of protein was loaded onto a 4–20% polyacrylamide gel (Bio-Rad Laboratories, Hercules, CA, USA) and electrophoresed at 120 V for 1 h. Protein transfer onto a nitrocellulose membrane was performed using the iBlot Dry Blotting System (ThermoFisher Scientific, Rockford, IL, USA). The membranes were then blocked with 5% Nonfat Dry Milk (Cell Signaling Technology, Danvers, MA, USA) for 1 h. Primary antibodies (see Table 2) were diluted in 0.1% TBS-Tween/5% BSA solution and incubated with the membranes overnight. The next day, membranes were incubated with secondary antibodies anti-mouse or anti-rabbit (Merck, St. Louis, MO, USA) with a dilution of 1/5000 for 1 h. Detection was carried out using the Novex™ ECL Chemiluminescent Substrate Reagent Kit (Invitrogen, Carlsbad, CA, USA), and images were captured with the Fusion FX Edge system (Vilber, Collégien, France).

### 2.9. Immunofluorescence Staining

Cancer cells or macrophages, seeded on coverslips in a 12-well plate, were fixed with 4% paraformaldehyde (Sigma-Aldrich, St. Louis, MI, USA) for 15 min and then rinsed with PBS. Subsequently, they were incubated with a blocking solution for 1 h, followed by a 24-hour incubation with primary antibodies (Table 3). For the p-STAT3 antibody, an additional cell membrane permeabilization step was performed using methanol at −20 °C for 10 min, followed by PBS washes.

The next day, cells were incubated with the secondary antibody (1/500 dilution) for 1 h, then rinsed with PBS and distilled water. Finally, cells were mounted on slides with Vectashield containing DAPI (Vector Laboratories, Newark, CA, USA) to stain the nuclei. The slides were then observed under a confocal microscope (Nikon Ti2 A1RHD25, Tokyo, Japan).

### 2.10. Transwell Assay

The migration assay was conducted using cell migration kits (24-well, 8 µm) purchased from Merck Company (Overijse, Belgium). Firstly, the chambers were hydrated with medium without FBS for 1 h at 37 °C. Medium supplemented with FBS was then added to each well. Cells were suspended in a medium without FBS and containing bufalin at IC_80_ concentration, then added to the center of the chambers. The plate was incubated at 37 °C for 48 h. After incubation, the medium was removed, and excess cells were gently wiped away with a cotton swab. The underside of the chambers, where cells had migrated, was fixed with 4% paraformaldehyde for 15 min and then stained with 4% crystal violet for 30 min. The chambers were washed with distilled water and visualized under a microscope. Brightfield images were captured using an inverted microscope coupled with a camera (EUROMEX HD). Quantification was performed by solubilizing the stained cells in Triton™ X-100 solution for 90 min, and the relative absorbance was determined at 570 nm, as described in a previous section.

### 2.11. Cell Cycle Assay

To measure the distribution of cells in cell cycle phases (G0/G1, S, G2/M), we used the Muse Cell Cycle Kit (Luminex, Austin, TX, USA). A cell suspension of 1 × 10^6^ cells was centrifuged, and the pellet was resuspended in 50 μL of PBS. The resuspended cells were then fixed in 70% iced ethanol while vortexing at medium speed. The suspension was frozen for at least 3 h at −20 °C before staining. The ethanol-fixed cells were centrifuged at 300× *g* for 5 min at room temperature. The cell pellet was then resuspended in 200 μL of Muse Cell Cycle Reagent and incubated for 30 min at room temperature, protected from light. The solution was vortexed and analyzed using the Muse flow cytometer (Guava^®^ Muse^®^ Cell Analyzer, Luminex, Darmstadt, Germany).

### 2.12. Statistical Analysis

The data shown represent three independent experiments (n = 3). Statistical analyses were conducted using IBM SPSS software (version 21) (IBM, Ehningen, Germany). Comparisons between independent experimental groups were made using either a *t*-test or an ANOVA, followed by Tukey’s post hoc test. A *p*-value of less than 0.05 was considered statistically significant (* = *p* < 0.05; ** = *p* < 0.01; *** = *p* < 0.001).

## 3. Results

### 3.1. Bufalin Inhibits the Proliferation of Head and Neck Cancer Cell Lines and Induced the Mitochondria-Mediated Apoptosis

First, the bufalin’s impact on cell proliferation was assessed in various cancer cell lines. The findings showed that bufalin significantly inhibited the proliferation of HNC cells at concentrations of 10 or 20 nM, depending on the specific cell line tested (Figure 1A). To further assess the relative cytotoxicity of bufalin on these three cell lines, IncuCyte cytotoxicity assays were conducted using IncuCyte^®^ Cytotox Green Dye to monitor cell membrane integrity disruption in real time and quantify cell death. The results revealed a significant increase in cell death in the treated cell lines compared to their controls (blue curves) (*p* < 0.01 and *p* < 0.001) (Figure 1B). Similar findings were observed in clonogenic growth analyses after 12 days of exposure to bufalin, showing that nanomolar concentrations of bufalin significantly reduced colony formation in the FaDu, 93-VU, and Detroit-562 cell lines in a dose-dependent manner (Figure 1C).

Next, we explored whether bufalin could induce apoptosis in these cell lines. A significant green signal, indicating apoptosis, was observed in FaDu and 93-VU cells compared to the negative controls (Figure 2A). Notably, the Detroit-562 cell line did not exhibit this fluorescence, suggesting a different cell death pathway. From these results, we choose to focus our subsequent experiments on the two cell lines most sensitive to bufalin-induced apoptosis.

To delve deeper into the molecular mechanisms behind these observations, we analyzed the protein levels of various apoptotic markers using Western blot analysis. In FaDu tumor cells, we found that levels of anti-apoptotic proteins Bcl-xL and Mcl-1 decreased, while levels of pro-apoptotic proteins Bax and cleaved caspase-3 increased (Figure 2B). Kinetic analysis of intrinsic pathway apoptotic markers revealed an increase in Apaf-1 after 3 h of treatment, a higher level of intracellular cytochrome c after 24 h, and an elevated amount of apoptosis-inducing factor (AIF) after 48 h. The 93VU cells showed different patterns for some proteins, but apoptosis was confirmed by a decrease in Mcl-1 and increases in Bax, Apaf-1, and cleaved caspase-3 (Figure 2B, the original Western blots have been shown in Appendix A). Additionally, TMRE labeling was conducted to assess changes in mitochondrial membrane potential. A significant decrease in TMRE expression was observed after 24 h of bufalin exposure in both FaDu and 93-VU cell lines (Figure 2C). These results align with the increased release of cytosolic cytochrome c, indicating that the mitochondrial apoptosis pathway plays a role in bufalin-induced cell death in HNC.

### 3.2. Bufalin Favors the Production of Reactive Oxygen Species by Downregulating Antioxidant Defenses

To determine if bufalin exposure induced oxidative stress, we measured mitochondrial ROS production in cultures treated with bufalin using the IncuCyte system over time. Mitochondrial dysfunction is often associated with oxidative stress. Utilizing Mito-SOX dye, we detected bufalin-induced superoxide anions in both cell lines. As shown in Figure 3A, bufalin treatment at IC_80_ significantly increased mitochondrial ROS levels compared to control cells (*p* < 0.001). This increase in superoxide anions was accompanied by a reduction in NRF2 expression, as evidenced by Western blot analysis after 24 h of treatment (Figure 3B, Appendix A). NRF2 is a transcription factor that regulates cellular antioxidant defense mechanisms by promoting the transcription of genes encoding antioxidant enzymes such as catalase and superoxide dismutase.

To further investigate the mechanisms of action of bufalin, we examined the effect of bufalin on the mRNA levels of these antioxidant enzymes using RT-qPCR analysis. The results showed significant downregulation of two major antioxidant enzymes, catalase and GSH reductase, in the FaDu and 93-VU cell lines. Specifically, catalase expression was significantly reduced in both cell lines (*p* < 0.05 and *p* < 0.001), while the reduction in GSH reductase expression was significant only in the 93-VU cell line (*p* < 0.05) (Figure 3C). Overall, these findings indicate that bufalin contributes to the creation of an oxidative and harmful environment for the tumor by disrupting mitochondrial function and decreasing the expression of key antioxidant defenses.

### 3.3. Bufalin Induces Cell Cycle Arrest by Disrupting Cyclin/cdk Complexes

We then examined the modulation of the cell cycle by bufalin by several approaches. The abundance of some markers involved in cell cycle progression or arrest was studied by Western blot after exposure to treatment at different times, using treatment kinetics. The results demonstrated a decrease in p53 along with a clear increase in p21 after 24 h and 48 h of treatment in FaDu cells. Due to the degradation of p53 by the E6 oncoprotein of HPV, no p53 nor p21 expression was observed in 93-VU cells (Figure 4A, Appendix A). Additionally, p21 upregulation was further confirmed at the protein and mRNA levels by immunofluorescence and RTqPCR analyses, as reported in Figure 4B,C, respectively. The relative mRNA expression of p21 was significantly enhanced after bufalin treatment for 24 h in both cell lines (*p* < 0.001, *p* < 0.01) (Figure 4C), while its nuclear protein expression was greater and more abundant, as observed by immunofluorescence in FaDu cells (Figure 4B). Regarding cyclin D1, we observed a reduction in its abundance from 3 h of treatment in both the FaDu and 93VU lines. Moreover, the kinase Cdk1 was also diminished over time compared to the untreated condition, only in FaDu cells (Figure 4A). Cell cycle distribution was finally investigated using the Muse Cell Cycle kit based on the quantification of propidium iodide in cells. The results showed a significant G2/M accumulation of 93-VU treated cells compared to untreated ones (*p* < 0.05) (Figure 4D). These results suggest that bufalin caused G2/M cell cycle arrest, which could potentially lead to cell death.

### 3.4. Bufalin Reduced Migration of Head and Neck Cancer Cell Lines by Regulating Epithelial to Mesenchymal Transition Markers

Because previous studies have suggested the potential of bufalin to inhibit migration and invasion in various cancers [19,42], we conducted a cell migration assay using transwell chambers on bufalin-treated FaDu cells over a 48-hour period. Our results indicated that bufalin significantly reduced cell mobility compared to untreated cells in Boyden chambers (Figure 5A,B). Additionally, we observed that bufalin treatment resulted in decreased mRNA levels of vimentin and Twist1, along with a significant increase in E-cadherin expression compared to untreated cells (Figure 5C). These findings imply that bufalin may influence cancer cell migration by modulating the epithelial–mesenchymal transition (EMT), enhancing their epithelial phenotype.

### 3.5. Bufalin Acts on EGFR and Affects Downstream Signaling Pathways Related to β-Catenin

Given that tumor progression in HNC cancers is predominantly governed by EGFR and the downstream signaling pathways mediated by this receptor, we hypothesized that bufalin might influence tumor proliferation and progression by acting on such receptor. To this end, we determined the effect of our molecule on the expression level of EGFR and important players involved in intracellular signaling by Western blot after treatment of FaDu and 93VU cell lines over time. After 6 h of treatment, we observed a decrease in EGFR abundance in the two cell lines, and this effect was even more marked after 24 and 48 h of treatment (Figure 6A). This observation was confirmed by immunofluorescence, where the results illustrated a sharp decrease in EGFR membrane expression (Figure 6C).

We then assessed how this effect could be transmitted downstream. As it is well known that β-catenin is commonly overexpressed in head and neck cancers and associated with a poor prognosis in patients, we sought to determine if treatment with bufalin could affect this overexpression. It was observed that in FaDu cells, the phosphorylated active form of β-catenin significantly decreased after 1 h of treatment, and this downregulation persisted up to 48 h. In contrast, in 93 vu cells, the decrease was observed only after 48 h (Figure 6A). Regarding phospho-STAT3, we obtained opposite effects between the two cell lines. As expected, a decrease in the absence of pSTAT3 expression was observed in FaDu cells after 3 h of treatment, while conversely, the protein was overexpressed in 93VU cells compared to the control condition after 6 h of treatment and beyond (Figure 6A, Appendix A). The absence of nuclear labeling in FaDu was confirmed by immunofluorescence after 48 h, supporting the cessation of this transcription factor’s activity following bufalin treatment (Figure 6C). 

In addition, the proto-oncogene and transcription factor c-Myc is an important regulator in cancer. Interestingly, we demonstrated that a 24-hour treatment with bufalin was able to drastically reduce the expression and activity of this protein, as observed by Western blot and immunofluorescence in both lines (Figure 6B,C). These results pave the way for further investigations into the mechanisms of action of bufalin in HNC.

### 3.6. Bufalin Drives M2 Macrophages toward the M1 Phenotype

We finally decided to focus our investigation on the immune response, especially on the regulation of bufalin on macrophage polarization. Firstly, we assessed the expression of markers specific to the two opposite types of macrophages, CD206 being related to the M2 pro-tumoral phenotype and CD86 to the M1 antitumoral phenotype. Monocytes isolated from the peripheral blood of healthy donors were purified, isolated, and polarized into M2 by the action of interleukins 4 and 13 for 48 h. The resulting macrophages were then incubated with 40 nM bufalin for 48 h and specifically identified by immunofluorescence. Our results confirmed a phenotypic reprogrammation of M2 macrophages toward an M1 pro-inflammatory phenotype. Figure 7A demonstrates a reduced expression of CD206 along with a stronger CD86 labeling after treatment. Moreover, we performed an immunolabeling of NFκB and revealed an enhanced expression of the protein as well as a relocation from the cytoplasm to the nucleus, suggesting a potential involvement of this transcription factor in the bufalin-induced repolarization (Figure 7A). In addition, bufalin significantly impaired the transcription of numerous cytokines, as observed in Figure 7B. Indeed, it remarkably decreased mRNA expression of CD206, CD163, IL-10, and MIF (*p* < 0.001) and increased the pro-inflammatory markers, including CD86, IL-12, iNOS, and NFκB (*p* < 0.05; *p* < 0.01; *p* < 0.001) (Figure 7B). These findings reinforce the role of bufalin in governing antitumoral immunity.

## 4. Discussion

Head and neck cancers are highly aggressive, leading to significant morbidity and mortality. The standard treatments for these cancers are generally associated with high toxicity and severe side effects adversely impacting patients’ quality of life and overall treatment outcomes [2]. Thus, there is an urgent need for new therapeutic strategies that are more effective with a lower toxicity profile.

Numerous studies have demonstrated that cardiotonic steroids like the ouabain possess anticancer properties across various types of cancer [24]. Bufalin, also a cardiotonic steroid, has shown anticancer properties as well. However, there is still a substantial gap in understanding bufalin’s effects on HNC cells. Therefore, our study aimed to address this lack by thoroughly analyzing bufalin’s mechanisms of action in HNC, positioning us as pioneers in this field of research. 

We first investigated the effects of bufalin on the proliferation of HPV+ and HPV- HNC cells and found that bufalin reduced cell viability in a dose-dependent manner, disrupted cell membrane integrity, and decreased colony formation. Secondly, bufalin induced apoptosis in FaDu and 93VU cells but not in Detroit cells. It also affected apoptosis-related protein levels, reduced mitochondrial membrane potential, and increased ROS production. Additionally, bufalin caused cell cycle arrest at the G2/M phase, reduced cell mobility, and altered EMT markers expression, highlighting its potential as a therapeutic agent for HNC. The observation that bufalin did not induce apoptosis in Detroit cells, an HPV-negative HNC cell line, implies that bufalin’s apoptotic effects may depend on other specific characteristics of different HNC cell lines. This selective response emphasizes the complexity of cancer cell mechanisms and suggests that bufalin may trigger apoptosis only under certain cellular conditions or genetic profiles. Variations in the expression of oncogenes, tumor suppressors, signaling pathways, and specific mutations could influence how cells respond to bufalin treatment. Investigating why Detroit cells resist bufalin-induced apoptosis could provide valuable insights into the molecular mechanisms underlying sensitivity or resistance to bufalin.

Given these findings, we explored the underlying pathways through which bufalin exerts these effects in FaDu and 93VU cells only. We initiated this by examining bufalin’s impact on the activation of phosphorylated β-catenin. Indeed, the Wnt/β-catenin signaling pathway promotes cell cycle progression, differentiation, migration, and tumor cell proliferation [13,43]. Increased nuclear levels of β-catenin are associated with tumor invasion and metastasis in oral cancers, as well as poor prognosis [44]. Our findings demonstrated that bufalin inhibited the phosphorylation of β-catenin at serine 675, a site known for activating β-catenin’s biological function [45]. Specifically, in FaDu cells, this reduction occurred within 1 h of treatment, while in 93VU cells, it occurred after 48 h.

The cell cycle-related kinase (CCRK) is an oncogenic regulator that facilitates β-catenin activation and nuclear translocation. Once in the nucleus, β-catenin forms a complex with the transcription factor TCF, which binds to promoter regions of genes such as EGFR and cyclin D1, thereby activating their expression [38]. Our experiments revealed that bufalin reduced EGFR expression after 24 h of treatment in both cell lines.

EGFR is overexpressed in 90% of HNC cases and is associated with poor overall survival and progression-free survival [46]. Elevated EGFR levels activate several downstream signaling pathways, including STAT3, which is hyperactivated in these cancers [46] and associated with increased cell proliferation and poor prognosis [47]. Our investigation revealed that the phosphorylation of STAT3 (Tyr 705) was decreased by bufalin after 3 h in FaDu cells, whereas it increased in 93VU cells.

Upon activation, phosphorylated STAT3 translocates into the nucleus and promotes the expression of target genes such as cyclin D1, survivin, Bcl-xL, and MCL-1, which are involved in tumor cell proliferation, angiogenesis, and immune evasion [13,48,49]. The balance between Bax and Bcl-xL proteins influences mitochondrial membrane potential (ΔΨm) [50,51]. This process leads to the release of cytochrome c into the cytosol, initiating apoptosome formation with Apaf-1 and activating caspase-9, followed by the recruitment and activation of caspases-3 and -7, which ultimately execute apoptosis. Furthermore, released apoptotic factors such as AIF and EndoG promote chromatin condensation and DNA fragmentation, driving programmed cell death [27,51]. Consequently, the inhibition of EGFR by bufalin in FaDu cells is hypothesized to reduce STAT3 phosphorylation and subsequently decrease the expression of these target genes encoding anti-apoptotic proteins, thereby initiating apoptosis through the mitochondrial pathway (Figure 8). Bufalin-induced apoptosis via the mitochondria-dependent pathway has also been documented in bladder [27], oral [28], and osteosarcoma cancers [29].

Interestingly, in HPV-positive cells, bufalin did not affect STAT3 activation. It is noted that HPV-associated cancers often exhibit elevated STAT3 activity [49]. It has been shown that STAT3 phosphorylation is significantly higher in HPV-positive cervical cancers compared to their HPV-negative counterparts, highlighting the role of HPV in enhancing STAT3 activity [52]. This suggests that HPV-related mechanisms may contribute to STAT3 activation despite bufalin treatment.

Furthermore, our investigation into bufalin’s effect on oxidative stress revealed an increased production of reactive oxygen species (ROS), primarily from mitochondria. High mitochondrial ROS levels trigger intrinsic apoptosis by inducing mitochondrial membrane pore formation, cytochrome c release, mitochondrial dysfunction, and cellular apoptosis [53,54]. Thus, bufalin induces apoptosis in cancer cells through the mitochondrial pathway by enhancing ROS levels. However, excessive ROS production can lead to cell death in both tumor and healthy cells [55]. In response, cancer cells often upregulate antioxidant enzymes to counteract oxidative stress. Our study demonstrated that bufalin reduces levels of NRF2, a transcription factor critical for antioxidant responses in cancer cells [56]. This effect on NRF2 suppresses the expression of key antioxidant enzymes such as catalase and GSH reductase, which are typically activated under oxidative stress conditions [57]. This dual mechanism underscores bufalin’s potential as a therapeutic agent by disrupting cellular antioxidant defenses and enhancing ROS-mediated apoptosis in cancer cells.

Concerning EMT, it has been shown that the β-catenin pathway plays a crucial role in this process, which sustains tumor migration and invasion. During EMT, a down-regulation of E-cadherin occurs along with an up-regulation of N-cadherin and vimentin [58]. Dysregulated β-catenin expression disrupts complexes formed with E-cadherin, leading to a loss of epithelial characteristics and the promotion of mesenchymal phenotypes. Bufalin has also been shown to disrupt the nuclear transport of β-catenin in colorectal cancer cells and, therefore, inhibit EMT [38]. The induction of EMT in FaDu cells by bufalin suggests that it achieves this effect through β-catenin inhibition. These findings underscore bufalin’s significant antitumor activity by suppressing cell migration and invasion. Similar effects have been documented in hepatocellular carcinoma cells. Additionally, bufalin reduced the expression levels of key transcription factors involved in EMT, namely Snail, Slug, Twist, and Zeb1 [35,37].

Moreover, the Wnt/β-catenin pathway is also known for its role in upregulating the proto-oncogene c-Myc [59], which functions as a transcription factor regulating genes involved in cell proliferation, differentiation and apoptosis [60,61]. c-Myc also promotes cell cycle progression by regulating cyclins (D, E, A, and B1) and CDKs (1, 2, 4, and 6) while suppressing cell cycle inhibitors such as p15, p21, and p27 [62]. Our study demonstrated that bufalin effectively reduced c-Myc expression in both tested cell lines. A previous study reported that bufalin affected a regulatory loop between c-MYC and EIF4A, leading to the inhibition of the c-Myc translation [63]. Additionally, bufalin decreased cyclin D1 expression, which is crucial for the G1/S phase transition, while increasing p21 expression. Notably, p21 acts as a checkpoint inhibitor that blocks the formation of the CDK1/cyclin B complex essential for the transition from the G2 to M phase in the cell cycle. Overexpression of cyclin D1 in HNC is associated with the progression of dysplastic lesions into in situ carcinoma and poor clinical prognosis [64]. These findings suggest that bufalin induces cell cycle arrest by inhibiting c-Myc expression, thereby preventing cyclin D1 activation and promoting p21-mediated cell cycle inhibition (Figure 8). Similar effects have been observed in other studies where bufalin induced G2/M arrest in gastric cancer [30], oral cancer [28], leukemia [31], bladder carcinoma, and colorectal cancer cells [33,34].

Our study also aimed to investigate the impact of bufalin on the immune response, particularly focusing on macrophage polarization. Our previous research indicated that the immunosuppressive microenvironment in HNC strongly promotes the polarization of macrophages toward the M2 phenotype, which predominates in the TME and contributes to tumor progression [41]. Therefore, targeting M2 macrophages in the TME represents a promising strategy for HNC treatment.

Our study demonstrated that bufalin reduced M2 markers and related anti-inflammatory cytokines in M2-induced macrophages while enhancing M1-associated markers and immunostimulatory cytokines. These findings underscore bufalin’s immunomodulatory effects, favoring the differentiation of M2 macrophages toward the M1 phenotype. To elucidate the mechanisms behind this reprogramming, further analyses were conducted. Our findings revealed that bufalin activates the NF-κB p65 subunit and inhibits the expression of macrophage inhibitory factor (MIF).

NF-κB is a master regulator of inflammation and immunity. It regulates macrophage functions by influencing the production of inflammatory cytokines. Activation of NF-κB has been shown to be responsible for the polarization of M2 macrophages into the M1 phenotype [39]. NF-κB activation is triggered by the phosphorylation of its inhibitor, IκB, in response to external signals such as cytokines. This allows for the release of the p65-p50 complex and its entry into the nucleus, promoting the production of immunostimulatory cytokines. Conversely, the p50 homodimer acts as a repressor, blocking NF-κB activity in the nucleus [65,66]. It has been demonstrated that bufalin specifically targets the p50 subunit of the NF-κB complex. This facilitates the entry of the p65-p50 complex into the nucleus while simultaneously reducing the formation of the p50-p50 complex and then activation to the NF-κB. Our results align with these findings, suggesting that bufalin induces polarization via NF-κB activation.

Additionally, the inhibition of MIF by bufalin led us to consider its potential impact on this pathway. MIF is a pleiotropic inflammatory cytokine involved in various cellular processes, including proliferation, migration, as well as anti-apoptotic activities [67]. Overexpression of MIF in HNC is linked to a poor prognosis [68]. MIF may be a key factor regulating M2 macrophage polarization, and thus, regulating MIF expression and release may mitigate M2 macrophage polarization. Studies have shown that bufalin could directly target the Steroid Receptor Coactivator-3 (SRC-3) protein to inhibit the proliferation of breast cancer cells [69]. SRC-3 is an oncogene involved in the development of various cancers through its role as a coactivator of transcription factors. It has been shown that SRC3 could directly activate the transcription of MIF [70]. In research on colorectal cancer, bufalin was shown to target SRC-3 directly, resulting in reduced MIF expression and influencing the polarization of M2 macrophages [40]. These findings suggest that bufalin may also inhibit this pathway in macrophages and induce reprogramming.

While our findings show bufalin’s multifaceted effects on HNC through its modulation of various cellular pathways, it is not yet clear if bufalin initiates a specific regulatory pathway responsible for its anticancer properties.

HuaChansu, approved by China’s FDA, is recognized for its anticancer properties, such as inhibiting cell proliferation, inducing cell differentiation, promoting apoptosis, and preventing cancer angiogenesis [71]. These effects are primarily due to bufadienolides, including bufalin, resibufogenin, and cinobufagin, which have demonstrated significant anticancer activities [24].

There is considerable evidence supporting the efficacy of HuaChansu (Cinobufacini), particularly when used in combination with other treatments. Clinical trials and meta-analyses across various cancer types, including hepatocellular carcinoma, non-small cell lung cancer, breast cancer, gallbladder cancer, gastric cancer, and liver cancer, have shown promising results [72]. Combining Cinobufacini with chemotherapy has improved overall response rates, clinical benefit rates, and quality of life in breast and gastric cancers, indicating potential synergistic effects [73]. Studies on gallbladder and lung cancers further support its use with conventional chemotherapy, showing improved survival rates and reduced side effects [74].

To maximize the therapeutic potential of bufalin and minimize resistance, we propose a hypothesis for a multidrug therapy approach. This strategy would leverage bufalin’s known anticancer effects in combination with other chemotherapeutic agents, utilizing the synergistic effects often observed with multiple drug administrations [75]. One potential approach is to combine bufalin with Cetuximab, an FDA-approved monoclonal antibody targeting EGFR, widely used in treating head and neck cancers [48]. Cetuximab blocks the binding of epidermal growth factor to its receptor, inhibiting the EGFR signaling pathway critical for tumor growth and survival. Bufalin’s ability to reduce EGFR expression suggests a complementary mechanism, which could enhance tumor regression when used alongside Cetuximab. This dual targeting could lead to more effective tumor suppression and potentially overcome resistance mechanisms associated with monotherapy.

## 5. Conclusions

Altogether, bufalin exhibits multifaceted effects on HNC through its modulation of various cellular pathways. Our findings suggest that bufalin inhibits β-catenin activity, downregulates EGFR expression, and reduces phosphorylated STAT3 levels. These events lead to the transcriptional regulation of pro-apoptotic genes like BCL-2 and MCL-1, promoting apoptosis via the mitochondrial pathway. Moreover, bufalin inhibits β-catenin-mediated activation of c-Myc, leading to the activation of p21 and inhibition of cyclin D1, thus inducing cell cycle arrest. Furthermore, by suppressing β-catenin, bufalin reduces the activation of EMT, which is crucial for tumor cell migration and invasion (Figure 8). Additionally, bufalin demonstrates an immunomodulatory effect by promoting macrophage polarization toward the M1 phenotype, which enhances antitumor immune responses.

These insights underscore bufalin’s potential as a therapeutic agent in HNC by targeting multiple pathways involved in cancer cell aggressivity and macrophages. However, further studies are essential to validate the specific mechanisms adopted by bufalin in co-cultures as well as in animals and to optimize its therapeutic application in clinical settings.

## Figures and Tables

**Figure 1 cancers-16-02739-f001:**
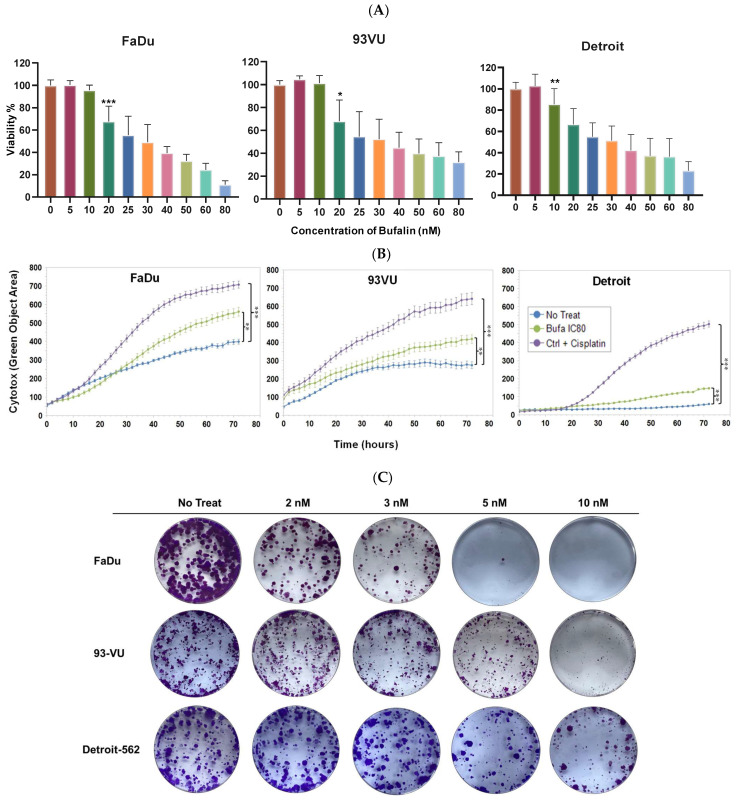
Bufalin’s effect on cancer cell proliferation. (**A**) Graphs showing cell viability percentages after treatment with varying concentrations of bufalin (0 to 80 nM). (**B**) Graphs illustrating the Cytotox green objective area (µm^2^) over time for three conditions: untreated; bufalin-treated at IC_80_; and cisplatin-treated positive control. Mean ± SD, Anova One Way, and Tukey’s post hoc test (* *p* ≤ 0.05; ** *p* ≤ 0.01; *** *p* ≤ 0.001. (**C**) Clonogenic assays evaluating the effects of bufalin at 2, 3, 5, and 10 nM.

**Figure 2 cancers-16-02739-f002:**
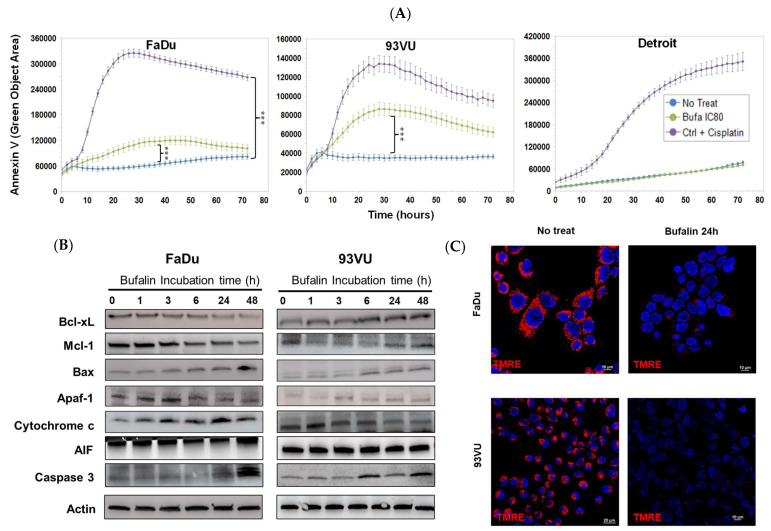
Bufalin’s effects on cancer cell apoptosis. (**A**) Graphs illustrating the Annexin V green objective area (µm^2^) over time for three conditions: untreated, bufalin-treated at IC_80_, and cisplatin-treated positive control at a concentration of 30 µM. Mean + SD, Anova One Way, and Tukey’s post hoc test (*** *p* ≤ 0.001). (**B**) Western blot analysis of apoptosis markers following bufalin incubation for various times, ranging from 1 h to 48 h, in both cell lines. (**C**) Immunofluorescence images of TMRE representing the ΔΨm before and after 24 h of bufalin treatment at IC_80_ (scale = 10 μm).

**Figure 3 cancers-16-02739-f003:**
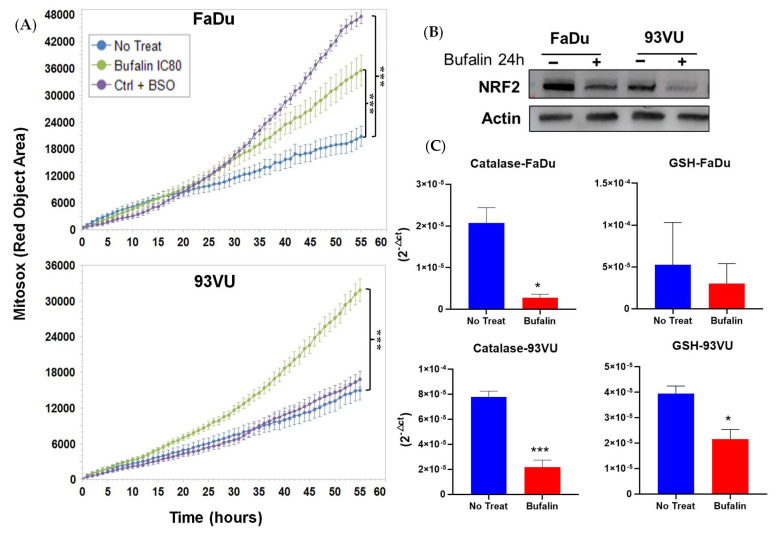
Bufalin’s effects on oxidative stress. (**A**) Graphs illustrating the Mitosox Red objective area (µm^2^) over time for three conditions: untreated; bufalin-treated at IC_80_; and BSO-treated positive control. Mean + SD, Anova One Way, and Tukey’s post hoc (*** *p* ≤ 0.001). (**B**) Western blot analysis of NRF2 marker following 24 h of bufalin IC_80_ incubation in both cell lines. (**C**) mRNA relative expression (2^−ΔCt^) of antioxidant enzymes before and after 24 h of bufalin treatment. Data are presented as Mean ± SD; *t*-test; * = *p* ≤ 0. 05; *** *p* ≤ 0.001.

**Figure 4 cancers-16-02739-f004:**
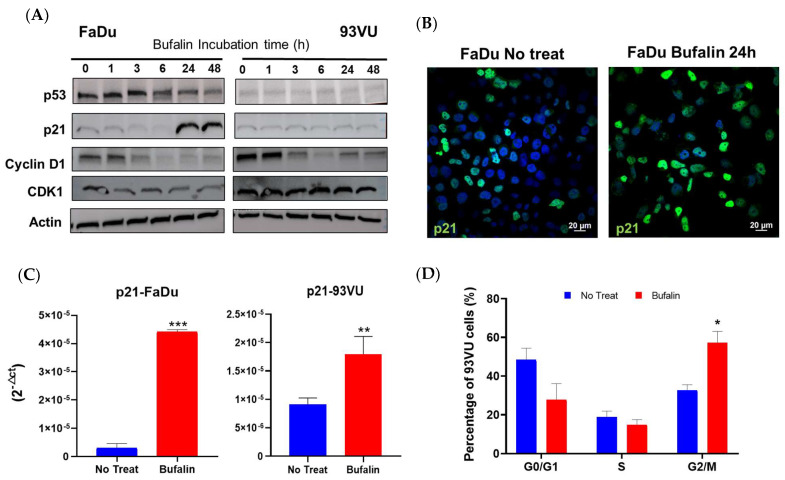
Bufalin effect on cell cycle regulation. (**A**) Western blot analysis of cell cycle-related markers following cell incubation with bufalin IC_80_ for various times, ranging from 1 h to 48 h. (**B**) Immunofluorescence confirming increased expression of p21 after bufalin treatment in FaDu cells. (**C**) mRNA relative expression (2^−ΔCt^) showing upregulated p21 following bufalin treatment. (**D**) Flow cytometry data representing the distribution of cells in different cell cycle phases after 24 h of bufalin treatment in 93VU cells. Data are presented as Mean + SD; *t*-test; * = *p* ≤ 0. 05; ** *p* ≤ 0.01; *** *p* ≤ 0.001.

**Figure 5 cancers-16-02739-f005:**
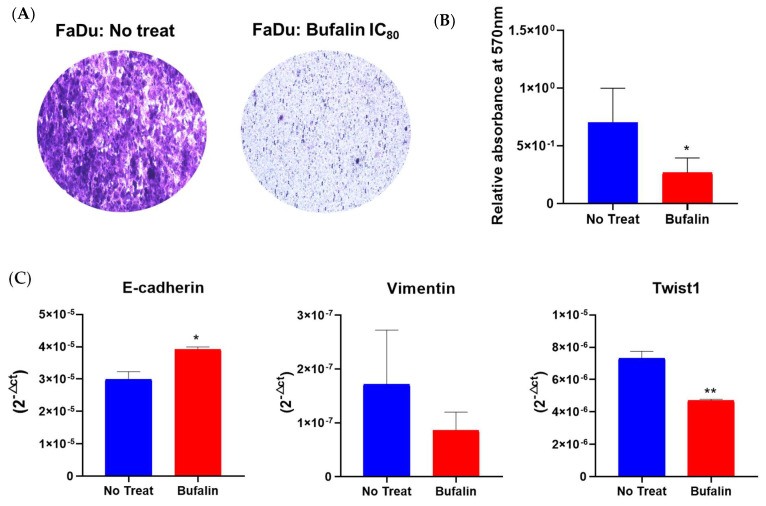
Bufalin effect on migration and EMT process. (**A**) Representative photographs of cell migration assays using transwell chambers after 48 h of bufalin treatment at IC_80_. (**B**) Quantification of migrated cells by measuring relative absorbance at 570 nm. (**C**) RT-qPCR analysis of mRNA expression levels (2^−ΔCt^) for EMT markers (E-cadherin, vimentin, Twist1), normalized to 18S expression, mean ± SD; *t*-test; * = *p* ≤ 0.05; ** = *p* ≤ 0.01.

**Figure 6 cancers-16-02739-f006:**
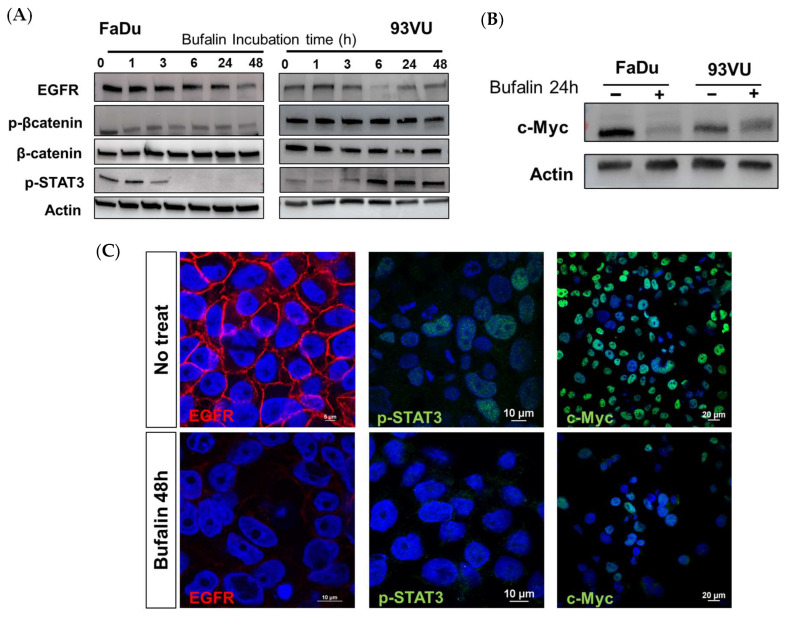
Bufalin downstream via different pathways. (**A**) Western blot analysis showing the expression levels of p-β-catenin (Ser675), total β-catenin, EGFR, and p-STAT3 (Tyr605) bufalin treatment for various durations, ranging from 1 to 48 h. (**B**) Western blot analysis of the proto-oncogene c-Myc marker following 24 h of bufalin incubation in both cell lines. (**C**) Immunofluorescence confirming decreased expression of EGFR (scale = 10 μm), p-STAT3 (scale = 10 μm), and c-Myc (scale = 20 μm) in FaDu cells after 48 h of bufalin treatment.

**Figure 7 cancers-16-02739-f007:**
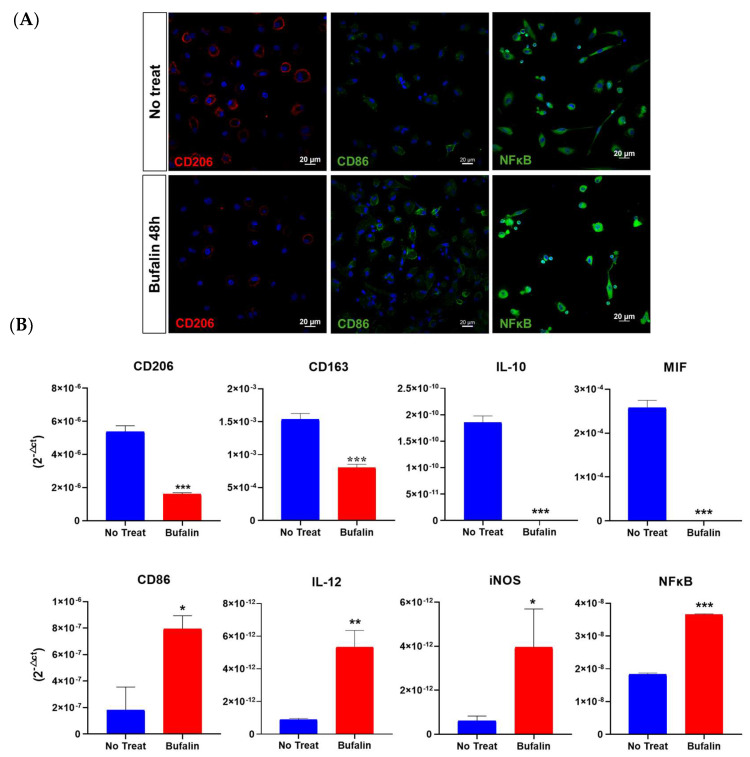
Bufalin effect (40 µM) on the reprogrammation of M2 toward M1 macrophages. (**A**) Evaluation of macrophage phenotypic markers CD86 and CD206 in M2-differentiated macrophages after 48 h of bufalin treatment by immunofluorescence (scale = 20 μm). (**B**) mRNA relative expression (RT-qPCR, 2^−ΔCt^) to characterize macrophage phenotype using M1 markers (CD86, IL-12, iNOS, NFκB) and M2 markers (CD206, CD163, IL10, and MIF). Mean ± SD, *t*-test (* = *p* ≤ 0.05; ** = *p* ≤ 0.01; *** = *p* ≤ 0.001).

**Figure 8 cancers-16-02739-f008:**
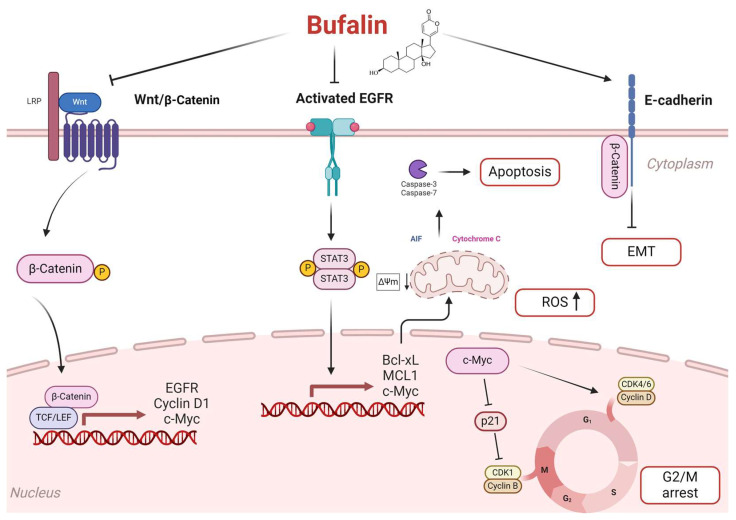
Proposal of the potential signaling pathways used by bufalin to induce apoptosis in HNC. Bufalin reduces β-catenin signaling activation by blocking p-β-catenin translocation to the nucleus and inhibiting the transcription of genes such as EGFR, Cyclin D1, and c-Myc. Inhibiting EGFR not only blocks the activation of the p-STAT3 pathway but also prevents the transcription of anti-apoptotic proteins like Bcl-xL and MCL-1. This inhibition shifts the balance toward pro-apoptotic factors, triggering apoptosis via the mitochondrial pathway. Consequently, cytochrome c and AIF are released into the cytosol, activating the caspase-3 cascade and initiating apoptosis. Additionally, the increase in reactive oxygen species (ROS) promotes apoptosis. Bufalin also blocks the cell cycle at the G2/M phase by inhibiting c-Myc. This action prevents the activation of cyclin D1 and promotes p21-mediated cell cycle arrest by blocking the formation of the CDK1/cyclin B complex. Finally, it prevents β-catenin nuclear translocation, preserving E-cadherin/β-catenin complex integrity and inhibiting epithelial–mesenchymal transition.

**Table 1 cancers-16-02739-t001:** List of qPCR primers.

Genes	Forward Sequences	Reverse Sequences
18S	CATTTAGGTGACACTATAGAAGACGATCAGATACCGTCGTAGTTCC	GGATCCTAATACGACTCACTATAGGCCTTTAAGTTTCAGCTTTGCAACC
Catalase	TTTTCGCCCTTAGCGTGAA	CATCCAGGTGAAAGCGGC
GSH Reductase	ACCCAACAGCCGCCGTAG	CAGACTGGTTGTTTCCATTCAGAT
IL10	TCAAGGCGCATGTGAACTCC	GATGTCAAACTCACTCATGGCT
CD206	CTACAAGGGATCGGGTTTATGGA	TTGGCATTGCCTAGTAGCGTA
E-cadherin	ATTTTTCCCTCGACACCCGAT	TCCCAGGCGTAGACCAAGA
Vimentin	AGTCCACTGAGTACCGGAGAC	CATTTCACGCATCTGGCGTTC
Twist1	CACTGAAAGGAAAGGCATCA	GGCCAGTTTGATCCCAGTAT
p21	TGTCCGTCAGAACCCATGC	AAAGTCGAAGTTCCATCGCTC
CD163	GGTGAATTTCTGCTCCATTCA	TGAGCCACACTGAAAAGGAA
MIF	GAACCGCTCCTACAGCAAG	AGTTGTTCCAGCCCACATTG
CD86	CTGCTCATCTATACACGGTTACC	GG AAACGTCGTACAGTTCTGTG
IL-12	GGGGAAGACCTGTGACTTGAG	AAAATAGATGCGTGCAAGAGAGG
iNOS	AGGGACAAGCC TACCCCTC	CTCATCTCCCGTCAGTTGGT
NFκB p65	ATCACTTCAATGGCCTCTGTGTAG	GAAATTCCTGATCCAGACAAAAAC
MIF	CATTTAGGTGACACTATAGAAGACGATCAGATACCGTCGTAGTTCC	GGATCCTAATACGACTCACTATAGGCCTTTAAGTTTCAGCTTTGCAACC
CD86	TTTTCGCCCTTAGCGTGAA	CATCCAGGTGAAAGCGGC
IL-12	ACCCAACAGCCGCCGTAG	CAGACTGGTTGTTTCCATTCAGAT
iNOS	TCAAGGCGCATGTGAACTCC	GATGTCAAACTCACTCATGGCT
NFκB p65	CTACAAGGGATCGGGTTTATGGA	TTGGCATTGCCTAGTAGCGTA

**Table 2 cancers-16-02739-t002:** Characteristics of antibodies used in Western blot.

Antibodies	Dilution	Molecular Weight	Firm	Species
Actin	1/1000	42	Sigma-Aldrich	Rabbit
Bax	1/500	20	Cell Signaling Technology	Rabbit
Bcl-xL	1/1000	30	Cell Signaling Technology	Rabbit
Mcl-1	1/1000	40	Cell Signaling Technology	Rabbit
Cytochrome c	1/800	15	Santa Cruz Biotechnology	Mouse
AIF	1/500	57	Santa Cruz Biotechnology	Mouse
APAF-1	1/500	130	Santa Cruz Biotechnology	Mouse
NF-κB p65	1/1000	65	Cell Signaling Technology	Rabbit
Cleaved Caspase 3	1/500	17	Cell Signaling Technology	Rabbit
NRF2	1/1000	97–100	Cell Signaling Technology	Rabbit
p21	1/1000	21	Cell Signaling Technology	Rabbit
p53	1/1000	53	Santa Cruz Biotechnology	Mouse
CDK1	1/2000	25–34	Proteintech	Rabbit
Cyclin D1	1/1000	37	Proteintech	Rabbit
EGFR	1/1000	175	Cell Signaling Technology	Rabbit
p-STAT3 (Tyr705)	1/1000	80	Cell Signaling Technology	Rabbit
p-β-catenin (Ser675)	1/1000	92	Cell Signaling Technology	Rabbit
β-catenin	1/1000	92	Cell Signaling Technology	Rabbit
c-Myc	1/1000	57–65	Cell Signaling Technology	Rabbit

**Table 3 cancers-16-02739-t003:** Table listing the different antibodies and their conditions used for immunofluorescence.

Targeted Proteins	Primary Antibodies	Dilution Ratios	Blocking Buffers	Secondary Antibodies
p21	Rabbit monoclonal anti-p21, Cell signaling (Danvers, MA, USA)	1/400	PBS with 1% BSA and 0.3% Triton	Goat anti-rabbit IgG (H + L), Alexa Fluor Plus 488
CD86	Rabbit monoclonal anti-CD86, Cell signaling	1/100	PBS with 5% normal goat serum (NGS) and 0.3% Triton	Goat anti-rabbit IgG (H + L), Alexa Fluor Plus 488
CD206	Mouse monoclonal anti-CD206, Cell signaling	1/100	PBS with 2% BSA	Goat anti-Mouse IgG (H + L), Alexa Fluor Plus 555
EGFR	Mouse monoclonal anti-EGFR, Thermo Fischer	1/200	PBS with 5% normal goat serum (NGS) and 0.3% Triton	Goat anti-Mouse IgG (H + L), Alexa Fluor Plus 555
p-STAT3 (Tyr705)	Rabbit monoclonal anti-p-STAT3, Cell signaling	1/100	PBS with 5% normal goat serum (NGS) and 0.3% Triton	Goat anti-Rabbit IgG (H + L), Alexa Fluor Plus 488
c-Myc	Rabbit monoclonal anti-c-Myc, Cell signaling	1/100	PBS with 5% normal goat serum (NGS) and 0.3% Triton	Goat anti-Rabbit IgG (H + L), Alexa Fluor Plus 488
NFκB p65	Rabbit monoclonal anti-NFκB p65, Cell signaling	1/800	PBS with 5% normal goat serum (NGS) and 0.3% Triton	Goat anti-Rabbit IgG (H + L), Alexa Fluor Plus 488

## Data Availability

All data are contained within the article.

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
