# Peer review of "Bufalin Suppresses Head and Neck Cancer Development by Modulating Immune Responses and Targeting the β-Catenin Signaling Pathway"

_cancers, 2024, doi:10.3390/cancers16152739_

Round 1

Reviewer 1 Report

Comments and Suggestions for Authors

The paper “Bufalin Suppresses Head and Neck Cancer Development by Modulating Immune Responses and Targeting the β-Catenin  Signaling Pathway” by Descamps et al deals with the promising activity, among others activities, of Bufalin against  head and neck cancer. It can be accepted for the publication in cancers after a major revision concerning the following points:

1)           Several different sentences should be shifted from Results (5) to Materials and Methods: i.e. Lines 255-256, Lines 259-261, Lines 275-276. Check the entire manuscript.

2)           Line 450, The authors should clearly explain the implications and the signficance of the fact that bufalin did not induced apoptosis in Detroit cells;

3)           The authors state: “…Thus, there is an urgent need for new therapeutic strategies that

are more effective with a lower toxicity profile….address this lack by thoroughly analyzing bufalin's mechanisms of action in HNC, positioning us as pioneers in this field of research…..”.

The authors show “…multifaceted effects on HNC through its modulation of various cellular pathways….” but is not clear if bufalin has started a regulatory pathway for the demonstrated anticancer properties nor is clear the therapeutic strategy although announced (Lines 440-441).

They should provide an hypothesis of therapeutic strategy aimed to minimize resistance and to maximize the well-known sinergic effect occurring when muliple drug are administered. They can hypothesize a multidrug therapy also administered through nanoparticles as glimpsed in the following paper DOI: 10.1002/smsc.202400113

4)           Too many undisclosed references.

The guidelines for reviewers  ask: “Are the statements and conclusions drawn coherent and supported by the listed citations?”

This condition cannot be observed if the reference is not clear, please show the source or delete the  undisclosed references.

Author Response

  • Several different sentences should be shifted from Results (5) to Materials and Methods: i.e. Lines 255-256, Lines 259-261, Lines 275-276. Check the entire manuscript.

Thank you for your comment. We have reviewed the entire manuscript and have relocated the specified sentences from the Results section to the Materials and Methods section (LINE 134-135, Line 159-161). Additional adjustments have been made as necessary.

  • Line 450, The authors should clearly explain the implications and the signficance of the fact that bufalin did not induced apoptosis in Detroit cells;

Thank you for your comment. We have added a paragraph on lines 464-472 that explains the implications and significance of bufalin not inducing apoptosis in Detroit cells, including a proposed hypothesis and future perspectives

  •   The authors state: “…Thus, there is an urgent need for new therapeutic strategies that are more effective with a lower toxicity profile….address this lack by thoroughly analyzing bufalin's mechanisms of action in HNC, positioning us as pioneers in this field of research…..”.

        The authors show “…multifaceted effects on HNC through its modulation of various cellular            pathways….” but is not clear if bufalin has started a regulatory pathway for the demonstrated anticancer properties nor is clear the therapeutic strategy although announced (Lines 440-441).

They should provide a hypothesis of therapeutic strategy aimed to minimize resistance and to maximize the well-known sinergic effect occurring when muliple drug are administered. They can hypothesize a multidrug therapy also administered through nanoparticles as glimpsed in the following paper DOI: 10.1002/smsc.202400113

Thank you for your comment. We have added a hypothesis of a therapeutic strategy aimed at minimizing resistance and maximizing the synergistic effects of multiple drugs. This addition can be found in lines 446-456.

  •    Too many undisclosed references.

The guidelines for reviewers ask: “Are the statements and conclusions drawn coherent and supported by the listed citations?”

This condition cannot be observed if the reference is not clear, please show the source or delete the undisclosed references.

Thank you for this remark. We have reviewed and ensured that all references are clearly linked to the corresponding statements in the text.

Reviewer 2 Report

Comments and Suggestions for Authors

The manuscript is dedicated to identifying the mechanisms of bufalin cytotoxicity, which was first described in the late 20th century. This compound and its structural analogues may be sought as components of combined therapy for malignant neoplasms. The manuscript makes a good impression. The set of methods is adequate and well described, the results are clearly presented and sufficiently discussed. However, at least in the discussion, the authors should have mentioned that other cardiotonic steroids (e.g. ouabain) are also capable of producing similar biological effects to bufalin. 

In addition to the above, minor errors should be corrected, such as:

1) Unit error (10 mg/mL (or 10 g/L) Streptomycin) - line 108;

2) It is necessary to give a reference to the basic descriptions of the methods used, or, if they are original, to indicate this. Or specify the name and manufacturer of the kits for Viability assay, Clonogenic assay;

3) Need to decode the abbreviation BSO - line 153

Author Response

  • Unit error (10 mg/mL (or 10 g/L) Streptomycin) - line 108;

Thank you for your remark. It has been corrected.

  • It is necessary to give a reference to the basic descriptions of the methods used, or, if they are original, to indicate this. Or specify the name and manufacturer of the kits for Viability assay, Clonogenic assay.

Thank you for your remark. A sentence was added at line 124.

3) Need to decode the abbreviation BSO - line 153

Thank you for your remark. It has been done.

Round 2

Reviewer 1 Report

Comments and Suggestions for Authors

The quality of the paper has been considerably improved but further improvements are requested for the publication in Cancer.

1)This reviewer believes that the glimpsed therapeutic use of bufalin still poorly supported with references: the authors should better represent and reference both the need for multidrug therapy approach and the technological route hypotesized in which bufalin is employed as part of a multi-targeted therapy.

2) Line 453, the authors should reference the following statement: “…utilizing the synergistic effects often observed with multiple drug administrations.”;

3) Line 454, The authors should detail, with respect to the kind of combination they imagine, and reference the following statement: “One potential approach is to combine bufalin with Cetuximab……”

Author Response

1)This reviewer believes that the glimpsed therapeutic use of bufalin still poorly supported with references: the authors should better represent and reference both the need for multidrug therapy approach and the technological route hypotesized in which bufalin is employed as part of a multi-targeted therapy.

We appreciate the reviewer’s feedback regarding the therapeutic use of bufalin and its representation in our manuscript. Here, we provide additional context and references to address these concerns.

While bufalin has not been used as a therapeutic approach alone or combined with other treatments, there is significant research demonstrating the efficacy of HuaChansu (Cinobufacini), a traditional Chinese medicine where bufalin is the principal bioactive component, when used in combination with other treatments for various types of cancer.

To better represent the need for a multidrug therapy approach and the hypothesized technological route for bufalin in a multi-targeted therapy, we have added a paragraph 589-603 describing the multitherapy approaches used by HuaChansu in combination with chemotherapy. This provides a perspective on the potential use of bufalin in combination with other therapies, supported by existing research on HuaChansu.

2) Line 453, the authors should reference the following statement: “…utilizing the synergistic effects often observed with multiple drug administrations.”;

Thank you. We added the reference that you proposed.

3) Line 454, The authors should detail, with respect to the kind of combination they imagine, and reference the following statement: “One potential approach is to combine bufalin with Cetuximab……”

Thank you for your comment.  Based on its impact on EGFR expression, we hypothesize that bufalin could be effectively combined with cetuximab, despite a lack of direct evidence in the literature to support this specific combination.

To address the reviewer’s request for more details on the kind of combination we imagine, we have added an explanatory phrase at lines 609-615. This addition highlights the rationale behind combining bufalin with cetuximab. Specifically, we propose that bufalin’s ability to reduce EGFR expression could enhance the efficacy of cetuximab, an FDA-approved monoclonal antibody targeting EGFR, which is widely used in treating head and neck cancers. This dual targeting could lead to more effective tumor suppression and potentially overcome resistance mechanisms associated with monotherapy.